

# Scrambling under quench

**Adith Sai Aramthottil**[1][⋆]**, Diptarka Das**[2][†]**, Suchetan Das**[2][‡] **and Bidyut Dey**[2][∘]

**1** Institute of Theoretical Physics, Jagiellonian University in Kraków,
Łojasiewicza 11, 30-348 Kraków, Poland
**2** Department of Physics, Indian Institute of Technology - Kanpur,
Kanpur 208016, India

⋆ adithsai.a@doctoral.uj.edu.pl , † didas@iitk.ac.in ,
‡ suchetan@iitk.ac.in , ∘ bidyutd@iitk.ac.in

## Abstract

We evaluate out of time ordered correlators in certain low dimensional quantum systems at zero temperature, subjected to homogenous quantum quenches. We find that when the Lyapunov exponent exists, it can be identified with the quenched energy. We show that the exponent naturally gets related to the post-quench effective temperature. In the context of sudden quenches the exponent is determined in terms of the quench amplitude while for smooth quenches we observe scalings (both the Kibble-Zurek as well as the *fast*) of the exponent with the quench rate. The scalings are identical to that of the energy generated during the quench.

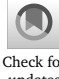

# 1   Introduction

An ongoing activity of considerable interest is to investigate the physics out of equilibrium. While on one hand is the question about the late time fate of the non-equilibrium quantum state, on the other hand we want to uncover features in the dynamical evolution of the state. Under suitable conditions (e.g. eigenstate thermalization) the system equilibriates to certain ensembles characterized by a temperature and/or chemical potentials conjugate to the conserved charges. When energy is the only conserved quantity one expects the ensemble to be a thermal one, characterized by a temperature; to which all observables finally equilibriate. It is believed that a thermalizing non-equilibrium state also *scrambles*. Here the notion of scrambling is in the context of operator growth due to chaotic dynamics. This is suitably measured by the Lyapunov exponent ($\lambda$) associated with the out of time ordered correlator (OTOC). In particular, for a maximally chaotic system in a thermal state, $\lambda$ is proportional to the temperature. For an out of equilibrium state that is thermalizing, we therefore expect the dynamical $\lambda$ to be related to the effective temperature to which this state equilibriates to. This temperature of course will depend on the energy generated during the quench, which further is governed by the particulars of the out of equilibrium scenario. Here we find evidences in some simple cases that the $\lambda$ evaluated in the non-equilibrium state shows aspects commensurate with energy generation, which can indeed be associated with thermalization in certain cases.

The non-equilibrium scenario is generated by the set up of quantum quench; wherein some coupling in the Hamiltonian is changed as a function of time. Various kinds of universalities are known to be associated with investigations of quenches. These emerge most notably whenever adiabaticity gets broken. This is guaranteed if the change in coupling is either sudden or if it crosses a critical point. Typically in a theory with cut-off $\Lambda$, the initial state is characterized by a mass gap $m_0$. The change of coupling $g(t)$ can be characterized by its rate of change $\Gamma$, and/or the change in the value of the coupling $\Delta g$.

For slow, smooth quenches in the regime, $\Gamma < m_0 < \Lambda$, the breakdown of adiabaticity is associated with the emergence of a Kibble-Zurek scale which imbues quenched correlation functions with universal scaling behaviours. There are also universal scalings expected in the fast regime, $\Lambda > \Gamma > m_0$ and in the sudden case $1/\Gamma \to 0$. See [1] for nice review on these two scalings. Evidences for these scaling behaviours have been limited to holography, solvable lattice models, two dimensional conformal field theories (CFTs), free theories and certain large $N$ theories. In this work, while remaining within this tractable set of illuminating theories we compute the OTOC during a quantum quench. Next, we outline the remaining sections along with the specific results from them.

In our first example we study the case of two harmonic oscillators $x$ and $y$ coupled by $gx^2y^2$ (see §2.1). This is a classically chaotic system, with classical Lyapunov index given by $\lambda_{cl} \sim E^c$, where $E$ is the classical energy. Interestingly, the quantum analog also exhibits a positive Lyapunov exponent $\lambda \sim T^c$ when the OTOC is computed in a thermal state with temperature $T$ [2]. The agreement with the classical exponent is not surprising as in the

thermal quantum case, the typical energy is given by the temperature. We consider quenching at zero temperature the coupling $g$, suddenly and compute the generated energy. Through the energy we are able to ascribe an effective temperature in terms of $\Delta g$, $T \sim (\Delta g)^\theta$. Next, by computing the OTOC, we extract $\lambda$ as a power of $\Delta g$ which is consistent with the assigned effective temperature. The computations are carried out numerically, and although bound to errors due to various truncations, clearly illustrate that the OTOC prognosticates an effective thermalization resulting from the quench. In this set-up this statement is, $\lambda \sim (\Delta g)^{c\theta}$.

The second example in §2.2 re-affirms the above observation of relating $\lambda$ and $\Delta g$ in a clean analytic example of sudden quench in 2D CFT in the limit of large central charge. We follow the Cardy-Calabrese prescription of approximating the quenched initial state with a regularized boundary state of the CFT. The regularization involves a parameter $\tau_0$ which is directly related to the mass gap of the initial theory. We then implement the three point BCFT set-up, in the semi-classical regime, to extract $\lambda = \pi/(2\tau_0)$. The parameter $\tau_0$ is consistently related to the initial mass gap as well as the effective temperature of the quenched state.

The final example in §2.3 deals with the Ising chain in presence of both a transverse as well as a longitudinal field. We quench the transverse field smoothly (with rate $\Gamma$) while operating in a non-integrable regime. Under both the fast as well as slow quench, the quenched energy is known to scale differently as a function of $\Gamma$. We employ tensor network techniques to extract $\lambda$ in this set-up and discover same scalings as for the energy generated during the quench.

The example of the coupled oscillator as well as that of the non-integrable Ising chain are examples with finite sized Hilbert spaces. Therefore several e-foldings of exponential growth as observed in chaotic semi-classical systems are absent. This has been pointed out for spin-chains with finite spin in [3], however the presence of *transient chaos* is not ruled out.[1] It is this transient regime where we find exponential commutator squared growth from which we extract a Lyapunov exponent. Our intention is not to connect this exponent to a classical definition, rather to provide evidence that the Lyapunov exponent associated with the squared commutator growth during transient regimes depends on the changing energy density during the quench process. All the three examples are indicative of the fact that in out of equilibrium scenarios where scrambling takes place, the Lyapunov exponent associated with suitable OTOCs can be identified with the energy generated. Furthermore, in scenarios where an effective post-quench temperature maybe estimated, the Lyapunov exponent is directly related with this temperature. This connection with thermalization emboldens us to use ETH for the commutator squared along with certain assumptions on the density of states, to further conjecture bound on the Lyapunov exponent extracted during quench as a function of the quench energy using techniques similar to [4]. In case of smooth quench with a characteristic rate, the energy density generated due to the quench is known to exhibit universal Kibble-Zurek scalings and the fast scalings [5,6]. We find that the exponent that we extract from the commutator squared growth during smooth quenches also follows exactly these same scaling exponents.

In case of the coupled quantum Harmonic oscillators, the exercise for smooth quenches is computationally very intensive. This will require us to solve for the instantaneous eigenfunctions and values by solving the Laplacian for every instant of time, each with a different coupling. This increases the computational complexity $T/\Delta T$ times in comparison with the sudden quench problem [which needs only the eigensystem of the initial and the final Hamiltonia], where $T$ is the time for which we evolve the system and $\Delta T$ is the timescales that we resolve numerically. Besides this technical hurdle, a main point that we elucidate with the oscillator example is the reflection of thermalization in scrambling. For this point, we do not need to quench smoothly. The sudden quench set-up is enough in this example to establish that the Lyapunov exponent is related to the quench energy. In the two dimensional CFT case, sud-

---

[1]In fact for the transverse field Ising model in the OTOC of observables that we consider this transient regime of exponential growth has been observed using the Pfaffian method, see §IV.B. of [8].

den quench was studied because unlike the Cady-Calabrese proposal there is no well-defined prescription to approximate states created by smooth quenches. A starting point will be to use conformal perturbation theory. For one point functions this has worked out in [6]. However for the OTOC computation this naturally leads to very high point functions (that need to be suitably integrated), which our bCFT set-up is specially designed to avoid, and will not lead to analytically tractable late time answers.

In the case of the Ising model, our focus was not on thermalization but on exhibiting the universal scalings associated with smooth quenches. Unlike the previous two examples, this model being amenable to sophisticated 1D numerical methods, allowed us to study smooth quenches: and indeed find both the Kibble-Zurek as well as the fast scalings.

## 2 Semiclassical chaos by differently ordered correlators

Classical chaos is quantified by exponential sensitivity to initial conditions, which is given by the following Poisson bracket (P.B.):

$$\frac{\delta x(t)}{\delta x(0)} = e^{\lambda_{cl} t} = \{x(t), p(0)\}_{P.B.}.$$

The generalization in quantum mechancis is via different time commutators [7],

$$C = \langle [V(t), W(0)]^2 \rangle,$$

where the square is taken in order to prevent phase cancellations. When the commutator is expanded we find out of time ordered correlators of the form, $F(t) = \langle V(t)W(0)V(t)W(0) \rangle$, make up $C$. As the quantum analog involves a square, we define the quantum Lyapunov via the exponential growth of $C \sim e^{2\lambda t}$. The exponent $\lambda$ has been shown to satisfy a universal bound set by the temperature when the expectation is evaluated in a thermal state. In our set-ups, instead of computing the OTOC in a thermal state we quench a zero temperature low lying quantum state of the initial Hamiltonian and extract the corresponding Lyapunov exponent.

### 2.1 Sudden quench of the coupled oscillator

For our first example we consider two coupled harmonic oscillators described by the Hamiltonian,

$$H = p_x^2 + p_y^2 + \frac{1}{4}(x^2 + y^2) + g x^2 y^2. \tag{1}$$

Note that we avoid the simpler linear coupling, $g\,x\,y$, as via a global unitary transformation ( in normal coordinates ) the linear coupling problem can be reduced to two decoupled oscillators. This makes the model integrable, and OTOC only shows oscillatory behaviour with time. This model has previously been explored in the OTOC context by [2,9]. While [2] investigated the transient chaotic regime, [9] discovered that the long time exponential behaviour of OTOC is restored in the classical limit. Our analysis focuses on the transient regime and in that sense is closer to [2].

Classically this has a non-zero $\lambda_{cl}$ which scales with energy as: $E^{1/4}$. In [2] the OTOC was computed in thermal state (temperature $T$) and it was found that $\lambda \sim T^c$ with $c \sim 0.25 - 0.31$. This is expected from the classical scaling since temperature plays the role of Energy. We compute the OTOC at zero temperature but under a sudden change of the coupling from $g_0$ to $g$. The quantity of interest is

$$C = -\langle n_0 | [x(t), p(0)]^2 | n_0 \rangle, \tag{2}$$

where $|n_0\rangle$ is a low lying energy eigenstate of the initial Hamiltonian $H(g_0) = H_0$. The time evolved position operator $x(t)$ is calculated with the quenched Hamiltonian $H(g)$: $x(t) = \exp(itH(g))x(0)\exp(-itH(g))$. We evaluate $C$ by inserting complete set of eigenstates of the new Hamiltonian $H(g)$ that we denote as $|m\rangle$.

$$C = -\sum_m \langle n_0|[x(t),p(0)]|m\rangle\langle m|[x(t),p(0)]|n_0\rangle = \sum_m b_m(t)b_m^*(t), \qquad (3)$$

the quantity $b_m(t) = -i\langle n_0|[x(t),p(0)]|m\rangle$ is Hermitian. Using another set of completeness insertions and the relation between matrix elements of $p$ and $x$: $p_{km} = \frac{i}{2}E_{km}x_{km}$ we find:

$$b_m(t) = \frac{1}{2}\sum_{l,k}\langle n_0|l\rangle x_{lk}x_{km}\left(E_{km}e^{iE_{lk}t} - E_{lk}e^{iE_{km}t}\right). \qquad (4)$$

Here, $x_{km} = \langle k|x|m\rangle$ and $E_{km} = E_k - E_m$. This is the main quantity that we compute numerically, which is plugged into eq(3) to get the OTOC. The wavefunctions $\langle x,y|n\rangle = \psi_n(x,y)$ are obtained by solving the Schrödinger's equation:

$$-\left(\frac{\partial^2}{\partial x^2} + \frac{\partial^2}{\partial y^2}\right)\psi_n(x,y) + \left(\frac{1}{4}(x^2+y^2) + gx^2y^2\right)\psi_n(x,y) = E_n\psi_n(x,y), \qquad (5)$$

with Dirichlet (wavefunction vanishing) boundary conditions in a square box.

### 2.1.1 Numerics

The equation (5) is solved for various values of the coupling $g$ that is relevant to us. The Mathematica® package NDEigensystem has been used to get the wavefunctions. The initial state $|n_0\rangle$ is taken to be a low lying eigenstate of $H_0$. The box size is kept at $20 \times 20$. The energy levels are suitably truncated in the completeness relations (see §A for justification of truncation). Once the wavefunctions $\psi_{n_0}(x,y)$ and $\psi_m(x,y)$ corresponding to $H_0$ and $H(g)$ respectively are available, we numerically integrate using the NIntegrate command to find the overlaps and the matrix elements as needed in (4). We show the behaviour of $C$ as a function of time for different changes in amplitude of the coupling, $\Delta g = g - g_0$, in Fig. 1. Also as can be seen in the figure, when the quench amplitude is very small then $C(t)$ does not have any exponential growth. All the plots have a regime where $C(t)$ grows exponentially with time, before beginning to oscillate in some complicated manner. Fortunately, the early time behaviour is sufficient to extract a Lyapunov exponent $\lambda$, since in any case due to truncation of high energy levels, late time numerics is unreliable. This is again with our expectation since the finite size of the local Hilbert spaces of coupled oscillator reduces scrambling window. Therefore, the OTOC $C(t)$ grows exponentially only in the transient time regime (see §A.1). To extract the Lyapunov exponent we fit $C(t)$ for different values of $\Delta g$ with $b\exp(2\lambda t)$. Next we plot $\lambda$ as a function of $\Delta g$ and find that $\lambda \sim (\Delta g)^\theta$. In Fig. 2, we present the case when the initial state is the $10^{th}$ eigenstate, for which the exponent turns out to be close to 0.25. For other initial states as well, we find numerical fits for the exponent to be close to a quarter. In the thermal ensemble $\lambda \sim T^c$ with $c$ extracted from a similar numerical regime and fitted to similar values. Therefore, this is encouraging as it suggests, that $\Delta g$ seems to play the role of temperature.

It turns out that there is further evidence for this "thermal" interpretation of $\Delta g$. We extracted an effective temperature for a given $\Delta g$ by equalizing energy computed in the thermal ensemble of the postquench Hamiltonian $H(g)$ and that in the quenched state, i.e. we solve for $T$ from the equation:

$$\langle H\rangle_T = \frac{\sum_k E_k e^{-E_k/T}}{\sum_j e^{-E_j/T}} = \sum_m E_m|\langle n_0|m\rangle|^2. \qquad (6)$$

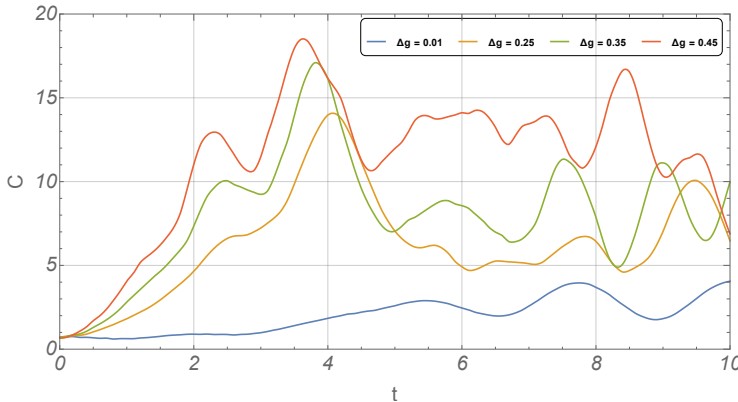

Figure 1: The time development of the commutator squared value shown for different quench amplitudes.

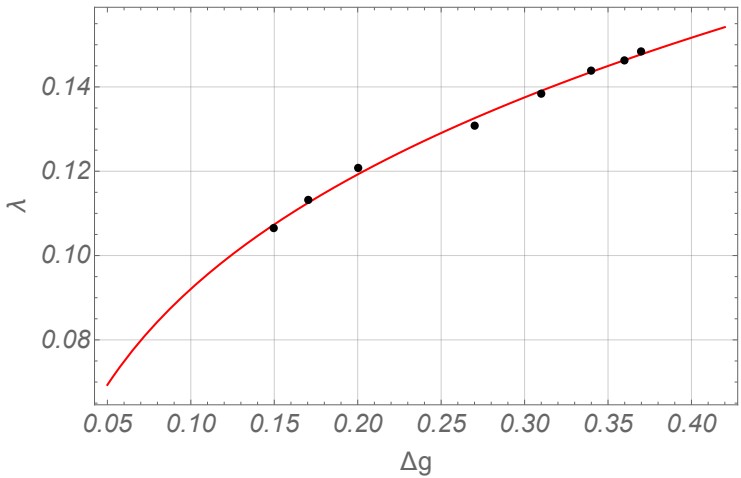

Figure 2: We fit the extracted $\lambda$ values (black) to $a + b(\Delta g)^{\theta}$ (red curve). When we restrict $\theta \in [0.1, 0.4]$ the best fit is obtained for $\theta = 0.255$. Upon including the variances, the coefficients lie in the following regions: $a \in [-0.04, 0.01]$, $c \in [0.2, 0.25]$ and $\theta \in [0.25, 0.33]$.

The R.H.S contains the information about the quench and uses numerical results that we already used to compute $b_m(t)$. In Fig. 3 we plot the effective temperature as a function of $\Delta g$ and obtain a straight line, this therefore strengthens our thermal explanation for the scaling of the Lyapunov exponent. We can also extract a temperature using the thermal ensemble result for the prequench Hamiltonian, in this case too we obtain a linear dependence on $\Delta g$.

## 2.2 Sudden quench in 2D CFT at large central charge

In this section we study the OTOC for sudden critical quenches in one spatial dimension. The initial state $|\psi_0\rangle$ is the ground state of a gapped Hamiltonian, with gap $\sim 1/\tau_0$. At time zero, the gap is closed suddenly as the Hamiltonian changes to $H_{CFT}$. From the perspective of the conformal theory, there is a boundary at Euclidean zero time, which is naturally associated with a boundary state of the CFT. Since in the infrared this state should be a conformal boundary state$|B\rangle$, a good approximation to $|\psi_0\rangle$ is provided by an irrelevant deformation to $|B\rangle$. The lowest universal irrelevant operator is a single power of the Hamiltonian itself. This is the

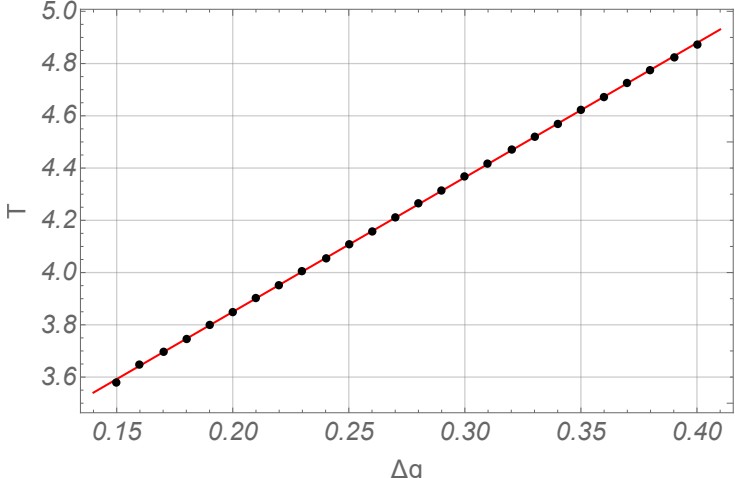

Figure 3: The effective temperature fits linearly with the quench amplitude.

proposed prescription of Cardy and Calabrese (CC) [10, 11]:

$$|\psi_0\rangle \propto e^{-\tau_0 H}|B\rangle. \tag{7}$$

Note, that the RG distance between $|\psi_0\rangle$ and $|B\rangle$, which is, $1/\text{gap} = \tau_0$ acts also as a regularizing parameter for the non-normalizable boundary state. Thereafter, post critical quench observables can be calculated as Euclidean strip (of width $2\tau_0$) correlators, which are suitable analytically continued to get the Lorentzian answers. The correlators of primary fields $\phi_i(x_i, \tau_i)(i = 1, 2, \ldots n)$ in the strip can be mapped to that on upper half plane(UHP) using the following conformal transformation

$$\omega \to z = i e^{\frac{\pi\omega}{2\tau_0}}, \ \omega = x + i\tau. \tag{8}$$

This mapping allows us to relate:

$$\langle\prod_{i=1}^{n} \phi_i(\omega_i, \bar{\omega}_i)\rangle_{\text{strip}} = \prod_{i=1}^{n} \omega_i'(z_i)^{-h_i} \bar{\omega}_i'(\bar{z}_i)^{-\bar{h}_i} \langle\prod_{i=1}^{n} \phi_i(z_i, \bar{z}_i)\rangle_{\text{UHP}}. \tag{9}$$

Here $(z, \bar{z})$ are UHP coordinates. Two or higher point functions are not fixed on the UHP, as using the conformal ward identities one can show that $n$-point BCFT correlators are equivalent to $2n$-point holomorphic CFT correlators on the entire plane. One point functions, however get fixed. An important one point function is the expectation value of the post-quench energy: $\langle\psi_0|H_{CFT}|\psi_0\rangle$. The contribution comes only from Schwarzian derivative associated with the map (8).

$$\langle B|e^{-\tau_0 H_{CFT}} H_{CFT} e^{-\tau_0 H_{CFT}}|B\rangle = \frac{\pi c}{96\tau_0^2}. \tag{10}$$

This result can also be obtained from the $T_{00}$ expectation value in a thermal ensemble with inverse temperature $\beta = 4\tau_0$, since

$$\frac{\text{tr}(H_{CFT} e^{-\beta H_{CFT}})}{\text{tr}(e^{-\beta H_{CFT}})} = \frac{\pi c}{6\beta^2}.$$

Hence, the quench gap / energy gets natually associated to an effective temperature.

**Extracting the Lyapunov exponent from BCFT**

In [12], the authors used a certain three point bulk-boundary thermal OTOC which shows maximal Lyapunov behaviour in large central charge limit of the BCFT. This is a suitable technical adaptation of the OTOC computation in 2D CFTs *without* any boundaries [13]. Here we consider three operators placed in the following way. $W$ of dimension $h_w$ is sitting at $(x,0)$ and two boundary operators $V$ of dimensions $h_v$ are sitting at $(0,t)$ in the Lorentzian $\omega$ coordinate. The OTOC can be obtained from the following object:

$$C(t) = \frac{\langle V(t)W(0)V(t)\rangle}{\langle W\rangle\langle VV\rangle}.\tag{11}$$

Operationally, to get the above out of time ordered object from the Euclidean correlator (with all operators $O_i$ at Euclidean time $\tau_i$ ), we analytically continue the $\tau_i$'s to their original Lorentzian values *i.e.* $\tau_i = t_i + i\epsilon_i$. Ultimately, we take all $\epsilon_i \to 0$ and the ordering in time comes from the ordering of the taking the limits of different $\epsilon_i$'s.

Using once again the map (8) the Euclidean correlator, $C_E$ is equivalent to a four point holomorphic function on the full plane, since only the bulk operator $W$ gets mirrored on the lower half plane. $C_E$ is just a function of the Euclidean cross-ratio $z$, and in the complex $z$ plane possesses branch-cuts originating from points when operators enter into each others' light cones. After analytic continuation, different time ordering dictates how or whether the branch cuts are being crossed. Here in our case, the four points on the plane, after the analytic continuation, are the following:

$$z_0 = ie^{b(x+i\epsilon_0)}, \quad \bar{z}_0 = -ie^{b(x-i\epsilon_0)}, \quad z_1 = ie^{b(t+i\epsilon_1)}, \quad z_2 = ie^{b(t+i\epsilon_2)}, \quad b = \frac{\pi}{2\tau_0}.\tag{12}$$

Here we take $\epsilon_1 = \tau_0$ and $\epsilon_2 = -\tau_0$ such that $z_1$ and $z_2$ are placed on the boundary of the UHP. Hence the cross ratio $z$ is given by $z = \frac{(z_0-\bar{z}_0)(z_1-z_2)}{(z_0-z_1)(\bar{z}_0-z_2)}$. For different time regimes, we get the following asymptotic behaviours for the cross-ratio:

$$z_{t\to 0} = \frac{2i\epsilon^*_{12}}{e^{bx}-e^{-bx}}, \qquad z_{t\gg x} = -2i\epsilon^*_{12}e^{-b(t-x)},$$

$$z_{t=x} = -\frac{2i\epsilon^*_{12}}{(1-e^{ib(\epsilon_0-\epsilon_1)})(1+e^{-ib(\epsilon_0+\epsilon_1)})} \approx \frac{\epsilon^*_{12}}{\epsilon^*_{10}}.\tag{13}$$

Here we have defined $\epsilon_{ij} \equiv i(e^{ib\epsilon_i} - e^{ib\epsilon_j})$ and $\epsilon^*_{ij}$ is the corresponding complex conjugate. From the above analysis, we see that the cross ratio goes to zero from opposite direction at $t \to 0$ and $t \to \infty$ limit and this is independent of the $\epsilon_i$ ordering. However at $t = x$, the cross ratio shows interesting behavior since it is a ratio of $\epsilon_{ij}$s. In particular for the OTOC, we need the following ordering: $\epsilon_1 > \epsilon_0 > \epsilon_2$. In this case, from (13) we could clearly see that $z_{t=x} \approx 1 + \frac{\epsilon^*_{02}}{\epsilon^*_{10}} > 1$. Diagrammatically the situation is described in the Fig. 4.

In the large central charge limit, we may assume that the dominant contribution to $C_E(z)$ comes from the Virasoro identitiy block, $\mathcal{F}(z)$ which in the monodromy regime of $h_i/c$ fixed with: $h_v \ll h_w$ has the universal form:

$$\mathcal{F}(z) \approx \left(\frac{z}{1-(1-z)^{1-12\frac{h_w}{c}}}\right)^{2h_v}.\tag{14}$$

To get to our desired OTOC, we need to wind around the branch cut at $z = 1$ and then as Lorentzian time increases, end up at small $|z|$. Therefore we implement $(1-z) \to (1-z)e^{2\pi i}$ and then expand in small $z$, which leads to:

$$\mathcal{F}(z) \approx \left(\frac{1}{1-\frac{24i\pi h_w}{cz}}\right)^{2h_v}.\tag{15}$$

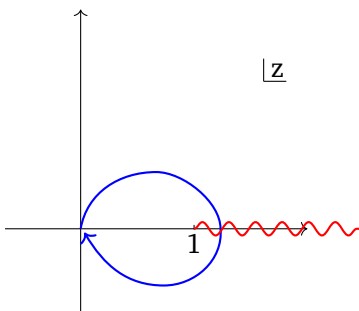

Figure 4: During the analytic continuation for OTOC the cut from 1 to infinity is crossed.

In terms of the Lorentzian time, $t$:

$$C(t) \approx \left( \frac{1}{1 + \frac{12\pi h_w}{\epsilon_{12}^*} e^{b(t-t_*-x)}} \right)^{2h_v}. \tag{16}$$

In the above expression, the scrambling time $t_* = \frac{1}{b} \log(c)$ and the Lyapunov exponent associated with $C(t)$ is

$$2\lambda = b = \frac{\pi}{2\tau_0}. \tag{17}$$

Comparing with the maximal Lyapunov exponent for the large $c$ thermal OTOC ($2\pi T$), we once again get the effective inverse temperature to be $\beta = 4\tau_0$. Once again, we see that the effective temperature associated with the quantum quench determines the scrambling behaviour. This result has also been obtained recently in [14].

## 2.3 Smooth quench in the quantum Ising model

We now explore the identification of the Lyapunov index with the energy generated during quench in the canonical quantum Ising chain. Intense quenching investigations has been carried out in this many-body lattice model. In particular, results exist for behaviour of the energy generated for smooth quenches. There are two distinct scaling regimes for smooth quenches, *viz.* the slow or the Kibble-Zurek (KZ) regime [15] and the fast regime [5].

KZ appears when the quench is *slow* compared to the initial mass gap, but still approaches criticality. In this case adiabaticity gets broken at a particular time. This sets a scale called the Kibble-Zurek time, $t_{KZ}$. The proposal of KZ is that for a window of time where adiabaticity is broken, all observables are frozen at this scale: Hence being close to the critical point one expects: $\langle O_\Delta \rangle \sim t_{KZ}^{-\Delta}$, where $\Delta$ is the scaling dimension of the observable $O_\Delta$ at criticality. There are also indications that this has been observed in experiments [15]. There is another limit of smooth quenches, where the quench is *fast* compared to the mass gap but smaller than the UV cut-off. In this case too there are universal scalings. These can be obtained from the underlying conformal field theory by using linear response, since the theory is insensitive to any scales other than the UV cut-off. Hence one expects: $\langle O_\Delta \rangle \sim \Gamma^{2-2\Delta}$. In the following we will see both these scalings realized for the *transient* Lyapunov exponent for the lattice model.

The quantum Ising model in one spatial dimensions is described by

$$H = -\sum_i J Z_i \otimes Z_{i+1} + g X_i + \epsilon Z_i. \tag{18}$$

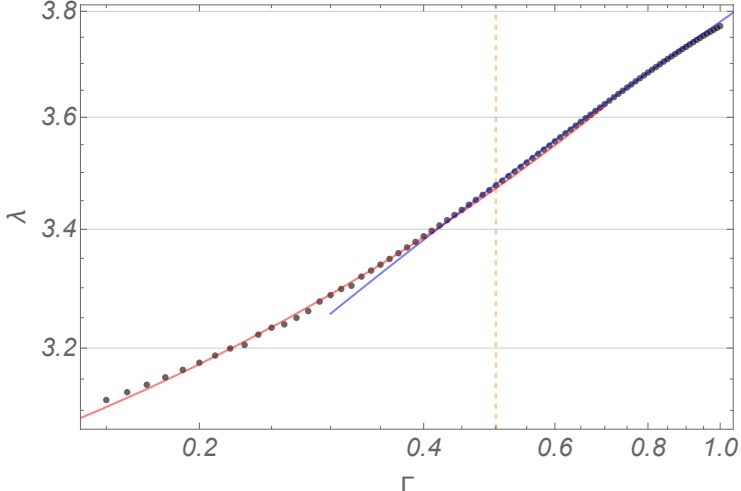

Figure 5: The Lyapunov exponent as a function of the quench rate $\Gamma$, shows $\sqrt{\Gamma}$ scaling (red is fit to $a_1 + a_2\sqrt{\Gamma}$) in the KZ regime which smoothly goes over to $\log\Gamma$ scaling (blue is fit to $a_1 + a_2\log\Gamma$) in the fast regime. The orange dashed line that demarcates the two regimes corresponds to $\Gamma = m = 1 - |g/J| = 0.5$.

We turn on both a transverse as well as an infinitesimal longitudinal field in order to be in the non-integrable regime. When the longitudinal field is absent, fermionization takes the model to a free massive ($m \propto 1 - |g/J|$) fermionic theory which is integrable. We quench linearly the coupling $g = g_0 + \Gamma t$ which plays the role of the transverse magnetic field. The Pauli $X$ is bilinear in terms of the Jordan-Wigner fermions, $\bar{\psi}\psi$ and has scaling dimension $\Delta = 1$, which is also the dimension of the energy operator. For relativistic theories in the KZ regime ($\Gamma < m$), the one-point functions $\sim t_{KZ}^{-\Delta}$. For the Ising model $t_{KZ} \sim \Gamma^{-1/2}$. Therefore the energy generated in the KZ regime is expected to show a $\sqrt{\Gamma}$ scaling. The observable $\langle O_\Delta \rangle$ is a time-dependent quantity, the scalings are best extracted when the measurement takes place at times $t$ such that for $t' \in (t - t_{KZ}, t + t_{KZ})$ there is a point for which $|g(t')/J| = 1$ i.e. the coupling crosses the critical point. This ensures that adiabaticity definitely gets broken.

The fast scaling regime is naturally defined by $\Lambda_{UV} = 1/(\text{lattice spacing}) > \Gamma > m$. Here we expect for the energy generated during quench to show: $\sim \Gamma^{2-2(\Delta=1)}$ scaling. Thus the energy generated shows logarithmic scaling with $\Gamma$. During smooth quenches of the transverse field Ising model, these scaling laws were showcased in the $\langle \bar{\psi}\psi \rangle$ operator expectation value [6].

Here, we compute the OTOC in the ground state of the initial $H(g_0)$:

$$C(t) = \langle \psi_0 | Z_j(t)Z_j(0)Z_j(t)Z_j(0) | \psi_0 \rangle. \tag{19}$$

Note, that we choose the $Z$ Pauli matrices since these are non-local in terms of the Jordan-Wigner fermions and have been shown (i) to exhibit "fast" thermalization and, even in the integrable regime, (ii) to mimic exponential Lyapunov behaviour in thermal state [8]. We find that the extracted Lyapunov exhibits the same smooth quench scalings as the energy generated, see Fig. 5. This once again strongly indicates that scrambling gets related to effective thermalization, and in fact that the energy generated via quench can be interpreted as the 'heat' generated [16, 17].

### 2.3.1 Numerical details

The OTOC has been evaluated with the help of tensor network (TN) techniques, see §Appendix B for further details. This allows us to go upto system size $L = 50$. We have chosen

periodic boundary conditions on the chain, and computed OTOC for Pauli $Z$ operators at the middle $= 25^{th}$ site. The parameters used for the numerics are: $g_0 = 0.5, J = 1, \epsilon = 0.0001$. Starting with the ground state in the ferromagnetic phase, we evolve the state with the time-dependent Hamiltonian, for different rates: $\dot{g} = \Gamma$. Numerically we implement this by discretizing the time-step in units of $\delta t = 0.01$. In the Schrödinger picture the OTOC can be represented as

$$OTOC(t) = \langle \psi_0 | Z_{25} \mathcal{U}(-t) Z_{25} \mathcal{U}(t) Z_{25} \mathcal{U}(-t) Z_{25} \mathcal{U}(t) | \psi_0 \rangle \,, \tag{20}$$

where the evolution operator is of the form $\mathcal{U}(t) = e^{-i\mathcal{H}(g(t))\delta t} \cdots e^{-i\mathcal{H}(g_0)\delta t}$. As described in the appendix since we use time-dependent variational principle algorithm (TDVP), the tensor network also gets optimized during the time-evolution. See Fig. 9 for a schematic of the contracted network used to extract the OTOC.

Since the OTOC is for Hermitian operators, we extract the squared commutator using: $C(t) = 2(1 - \Re(OTOC(t)))$. For the quenches we observe $C(t)$ to behave similar to that of the coupled oscillator quench Fig. 1, there is a clear transient period where the commutator squared grows exponentially before starting to oscillate, see Fig. 6. Fortunately for us, the OTOC characterization is for very early times, and thus we do not have to deal with the growing entanglement at late times which acts as a bottleneck for the TN techniques owing to finiteness of the bond dimension. We have checked that for the times of our interest, a bond dimension of $\chi = 32$ is enough for the extraction of $\lambda$. As shown in the figure, we extract $\lambda$ by fitting an exponential between $0.15 \leq t \leq 0.70$, for different rates. From this data we find the scaling shown in Fig. 5.

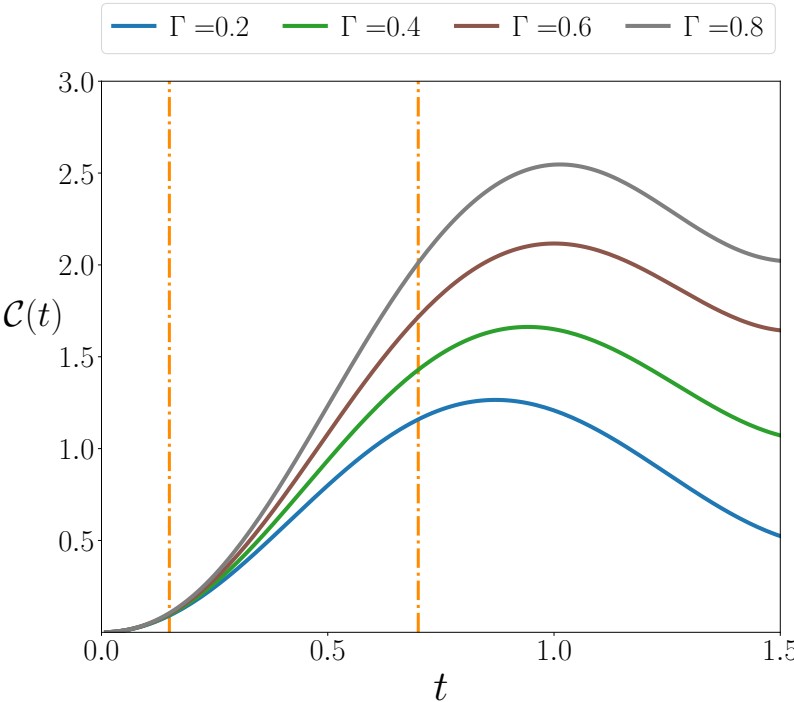

Figure 6: The time development of the commutator squared value is shown for different rates ($\Gamma$) of the smooth quench. The dashed-dotted lines represent the points in the plot with increasing ramps which we use to extract the Lyapunov exponent.

In theories like the Ising model, operators have bounded norms. Therefore only transient chaotic behaviour is observable, since the bounds get saturated very soon. This motivates the

consideration of density of OTOCs of extensive sums of local operators, which can show longer regimes of growth in time [18]. For the density observables one expects algebraic growths in time $\sim \alpha t^\beta$, from which one can once again obtain a scale which controls the rate of this growth

$$\tilde{\lambda} \;\; = \alpha^{1/\beta}\,. \tag{21}$$

It is for this quantity that we investigate the scaling behaviours, and find both the *fast* & the *Kibble-Zurek* regimes as observed in the Lyapunov exponent. See §Appendix C for further details.

## 3 A bound on the quenched energy dependence of $\lambda$ under sudden quench

From the examples of OTOC in quantum systems under quench, we have found that $\lambda$ is proportional to some positive power of the quenched energy $\epsilon$, *i.e.* $\lambda \propto \epsilon^\kappa$. We have also seen that often an effective temperature maybe ascribed to describe the quench. In that case the effective temperature is related to $\epsilon$. Furthermore, in the context of thermal states, under very generic assumptions of analyticity $\lambda$ satisfies a bound set by the temperature [19]. Therefore it is natural to postulate that $\kappa$ maybe bounded. We conjecture, that under a sudden quench: $H = H_0 \theta(-t) + H\theta(t)$, with $|\psi_0\rangle$ an initial eigenstate of $H_0$:

**Conjecture 1** *If $C(t) \equiv \langle\psi_0|[W(t), V(0)]^2|\psi_0\rangle \sim e^{2\lambda t}$ and $\epsilon \equiv \langle\psi_0|H|\psi_0\rangle$, then $\lambda \approx \epsilon^\kappa$ with $\kappa \leq 1$.*

Here we provide a simple argument along the lines of [4] subjected to the following approximations:

1. We consider the systems with the typical density of states at large energy $E$ as of the form: $\rho(E) = E^\gamma, \gamma > 0$. Of course, quantum systems like 2d CFT or string theory do not lie into this class *e.g.* for CFT$_2$, $\rho(E) \sim e^{2\pi \frac{c}{6}\sqrt{E}}$ when $E \gg c/12$.

2. We assume the eigenstates of post-quench Hamiltonian satisfy ETH [20, 21] *i.e.* the off-diagonal matrix elements of expectation value in energy eigenstates are exponentially suppressed in entropy. In particular this is assumed to also hold for $C(t)$: $\langle i|C(t)|j\rangle \ll \langle i|C(t)|i\rangle$ where $|i\rangle, |j\rangle$ are energy eigenstates of $H$. This assumption stems from the fact that $\lambda$ has a thermal description, and even may have a thermal origin [22].

3. We assume that the decomposition of the initial state $|\psi_0\rangle$, in terms of the final $H$ eigenstates is characterized by a smooth function, and that $|\psi_0\rangle$ has finite energy w.r.t $H$.

Let us denote the eigenbases of $H$ as $\{|l\rangle\}$ with energy eigenvalues $\{E_l\}$. Since this forms a complete set, we can expand $\psi_0$ as

$$|\psi_0\rangle = \sum_l c_l |l\rangle\,, \qquad \sum_l c_l^* c_l = 1\,. \tag{22}$$

Hence,

$$\epsilon = \langle\psi_0|H|\psi_0\rangle = \sum_l c_l^* c_l E_l\,. \tag{23}$$

In the continuum limit, we take $c_l^* c_l \to g(E)$, to be a smooth function, and $\rho(E) = \sum_l \delta(E - E_l)$. This yields

$$\epsilon = \int_0^\infty dE \; g(E)\rho(E)E, \qquad \int_0^\infty dE \; g(E)\rho(E) = 1. \qquad (24)$$

Next using the first and third assumptions, we see that $g(E)$ needs to fall off exponentially (power law is excluded from convergence requirements at both limits of (24)). A consistent behaviour for $g(E)$ is given by $\approx \epsilon^{-1-\gamma} \exp\left(-\frac{\Gamma(2+\gamma)}{\Gamma(1+\gamma)} E/\epsilon\right)$. Now we consider the commutator square $C(t)$

$$C(t) = \langle \psi_0 | [W(t), V(0)]^2 | \psi_0 \rangle \approx \sum_l c_l^* c_l \langle l | [W(t), V(0)]^2 | l \rangle, \qquad (25)$$

where in the last approximation we used the ETH assumption. Hence in the continuum limit we have

$$C(t) \sim \int_0^\infty dE \; g(E)\rho(E) \langle E | [W(t), V(0)]^2 | E \rangle$$
$$\sim \epsilon^{-1-\gamma} \int_0^\infty dE \; \rho(E) e^{-\frac{\Gamma(2+\gamma)}{\Gamma(1+\gamma)} E/\epsilon} \langle E | [W(t), V(0)]^2 | E \rangle. \qquad (26)$$

We can recognize the above expression with the thermal $C(t)|_\beta$ with $\beta \propto 1/\epsilon$. Thus we see the quenched energy $\epsilon$ sets the notion of effective temperature in this scenario. From MSS bound [19], we know the thermal Lyapunov is bounded by $\frac{2\pi}{\beta}$. Hence we get $\lambda \leq \alpha \, \epsilon$, for some constant $\alpha$. Therefore we have arrived at the bound $\kappa \leq 1$. From our analysis of critical sudden quench we have seen that for large $c$ 2d CFT, $\lambda \propto \sqrt{\epsilon}$. We can get the same bound with similar analysis as above if we considered the Cardy density of states $\rho(E) = e^{2\pi\sqrt{\frac{cE}{6}}}$ at high energy. This agrees with the bound in three point 'b-OTOC' in Cardy-Calabrese state in the similar spirit of deriving the MSS bound [14].

## 4 Conclusions

In this work we have studied scrambling during quantum quenches. All the chosen models start from a gapped low lying eigenstate of an initial Hamiltonian and then during the quench evolution either gets coupled to another system, or experiences sudden criticality, or evolves with a smoothly changing coupling. In our first example of coupled oscillators, we have studied growth of commutators during sudden quench. We showed that even in the transient regime, we can extract a Lyapunov exponent $\lambda$ which scales with quench amplitude $\Delta g$ in the same way as it scales with equilibrium temperature $T$. This further ascribe an effective temperature in terms of quench amplitude, i.e., $\Delta g \propto T$. In the 2d CFT, the sudden quench can be realized by the well known CC ansatz where the ground state of a gapped Hamiltonian with gap $\sim \frac{1}{\tau_0}$ is taken to be a conformal boundary state which is deformed by CFT Hamiltonian. In this set up, we have seen exponential behavior of three point bulk-boundary OTOC for large $c$ CFT from where we extract the Lyapunov exponent $\lambda$. Even in this infinite dimensional system, $\lambda$ is related to the energy generated during the quench as $\lambda = \frac{\pi}{2\tau_0}$ which sets an effective temperature $T = \frac{1}{4\tau_0}$. In these two examples, a central point we elucidate is the reflection of thermalization in scrambling, be it transient or regular. In the case of the Ising model, our focus was not on thermalization but on exhibiting the universal scalings associated with smooth quenches. We smoothly quenched the transverse field with a finite rate in the Ising

chain in the presence of a longitudinal field. The quenched energy scales differently with the rate in the fast and slow quench regimes. The extracted Lyapunov exponent exhibits the same smooth quench scalings with rate as the energy generated in both quench regimes.

In all the situations, for suitably chosen operators, signs of scrambling in the form of exponential OTOCs are observed. The extracted Lyapunov exponents have the characteristics of an effective temperature since it can be identified with the energy generated during the quench. Furthermore in the case of smooth quenches the extracted Lyapunov exhibits both a Kibble-Zurek as well as a fast scaling as expected in the quench energy.

The coupled oscillator studied in this work can be shown to arise from the dimensional reduction of $SU(2)$ Yang-Mills Higgs theory in the unitary gauge [2]. In the thermal context the scaling of the Lyapunov index with the temperature is expected to hold for the full Yang-Mills as well as its susy generalizations. A very interesting generalization is the $SU(N)$ Yang-Mills in $9 + 1$ dimensions with large $N$. When dimensionally reduced to zero dimensions, it is described by D0-brane matrix quantum mechanics. At high temperatures the Lyapunov index of this matrix model also shows the scaling $\lambda \sim T^{1/4}$ [23, 24]. It is then expected that the OTOC during a quench in these generalizations will also exhibit scalings. Using the gauge/gravity correspondence this will translate to universal scalings in genuinely non-equilibrium quantum gravity processes, which may include black hole formations as well as black hole transitions. It is to be noted, that equilibriation after quenches have already been studied in various matrix models [25].

As pointed out here as well as in [2], the exponent of 1/4 has its origin in the behaviour of the classical Lyapunov exponent. However the Lyapunov index extracted from the OTOC can be distinct from the classical Lyapunov exponent in certain physical systems as shown in [26, 27]. Hence, it will be quite interesting to carry out the quench and investigate the nature of effective thermalization as determined from scrambling in these models. Recently [22] has revealed a very interesting connection between classical Lyapunov and quantum tunneling resulting in a thermal distribution of *excess* energy. It will be very interesting to investigate if this connection continues to hold even for time-dependent set-ups.

In the context of the gauge/gravity duality, universal scalings of quenched correlation functions in the field theory can be understood from bulk zero mode dynamics in the dual gravity theory [28]. On the other hand, OTOCs in holography can be computed from the bulk using a shockwave set-up [29]. It will thereby be interesting to imbue the end of world brane geometry (dual to the Cardy-Calabrese quenched state [30]) with shockwaves and investigate the role of zero modes in the Lyapunov scalings. It is interesting to note, that in the context of thermal holographic quenches, the Lyapunov index jumps to the appropriate thermal value [31].

A common underlying theme in our examples is the emergence of a scale through an effective "thermalization". It is a natural question to ask what happens to the Lyapunovian behaviour when there are additional conserved charges? In this case one expects the final state to equilibrate to a generalized Gibbs ensemble, with generically non-zero chemical potentials turned on for different charges. For a CFT with an additional global $U(1)$ scrambling has been explored in the context of local quenches [32]. We expect that signs of such effective equilibration will get reflected in the scrambling characterizations associated with quenches.

When there are as many conserved charges as the degrees of freedom, a theory becomes integrable. Integrable theories are known not to show exponential OTOCs, however in presence of non-integrable perturbations can have a rich phase diagram of Lyapunovian dynamics, as was shown using the quantum tangent space formalism in [33]. Though our analysis of the Ising model falls in this class, it will be interesting to explore more generally within this framework the case of scrambling under quench.

In context of time-dependence and the quantum Ising model, OTOCs have also been explored in [34] and very recently in [35]. While the former establishes OTOCs as an order

parameter during sudden quenches. the latter shows how the information about dynamic phase transitions are present in the OTOC corresponding to non-local operators. Both these studies are based on late-time properties of the OTOC, whereas our investigation is complementary since it focuses on the early time behaviour. It will also be interesting to explore scrambling during smooth quenches in models with richer critical structures and also in presence of *disorder*, *e.g.*, [36].

Recently TN + Prony method has been used in the non-integrable Ising model at finite temperature to compute unequal time commutators [37] with some success. It seems only natural to adapt these techniques to explore non-equilibrium scrambling in future.

It will also be interesting to explore OTOCs in quenches for analytically controllable interacting field theories. At finite temperatures both in the $O(N)$ non-linear sigma model [38] as well as the Gross-Neveu model [39] the Lyapunov exponent scales linearly with temperature at large $N$. Furthermore, the Schwinger-Keldysh formalism for studying quantum quenches have already been explored in these models [40, 41]. It will be interesting to contrast the effective time-dependent temperature extracted from the Lyapunov index against that extracted from the time-dependent gap equation.

**Note Added:**

While this work was nearing completion, the recent preprint [14] has explored OTOCs during inhomogenous quenches in the context of two dimensional CFTs, wherein they used the BCFT answer (17), see Appendix A in [14].

## Acknowledgements

It is a pleasure to thank Titas Chanda, Amit Dutta, Bobby Ezhuthachan, Michal P. Heller, Arijit Kundu and Arnab Kundu for several useful discussions. We would also like to thank the anonymous second referee in SciPost for asking the very interesting question about the DOTOC which we explored in the present version. DD, SD, & BD would like to acknowledge the support provided by the Max Planck Partner Group grant MAXPLA/PHY/2018577. DD would also like to acknowledge the support provided by the MATRICS grant SERB/PHY/2020334. ASA would like to thank Titas Chanda for Tensor Network codes. ASA would like to acknowledge the support provided by National Science Centre (Poland) under project 2019/35/B/ST2/00034. BD would like to thank Supratim Das Bakshi for useful comments on `Mathematica` code. BD also acknowledges MHRD, India for Research Fellowship.

## A    Truncation of energy levels

In the calculation of the commutator squared or OTOC, $C(t)$ is given by equation (3) where we have used complete set of eigenstates of the new Hamiltonian $H(g)$ as

$$\sum_{n=1}^{\infty} |n\rangle\langle n| = 1.$$

Here $|m\rangle$ is the eigenstate of $H(g)$. However when we are calculating $C(t)$ numerically we truncate the sum over energy levels by a finite number $N$. To test numerical accuracy we look at the dependence of the OTOC on this truncated value. We plot $C(t)$ as function of time with fixed quench amplitude $\Delta g$ for different truncation levels $N$, see Fig. 7. For $N > 200$ the difference between OTOC is negligible since the plots for $N = 200$ and $N = 250$ are almost

coincident. On the other hand the plots for $N = 50$ and $N = 100$ do not coincide even in the early time regime $(0.5 - 2.5)$ where $C(t)$ is growing exponentially with time.

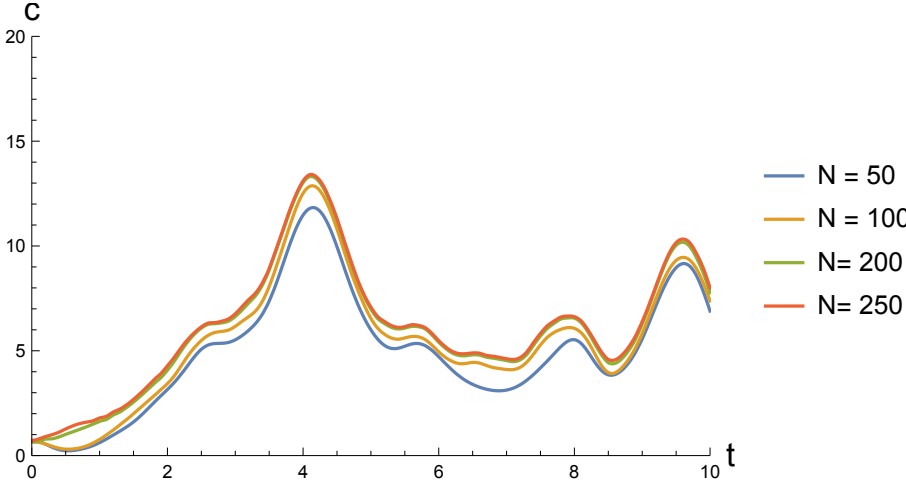

Figure 7: The time development of the commutator squared value shown for different truncation.

## A.1   Exponential growth of OTOCs with time

We present below in Fig. 8 the logarithmic plot of $C(t)$ with time in early $(0.75 - 2.4)$ time regime and the logarithmic plot clearly shows that there are straight lines. This is a clear indication of exponential growth of $C(t)$ in the transient time regime. This is the time regime where we have fit our data for $C(t)$ with $b\exp(2\lambda t)$ to extract $\lambda$ for different quench amplitudes $\Delta g$.

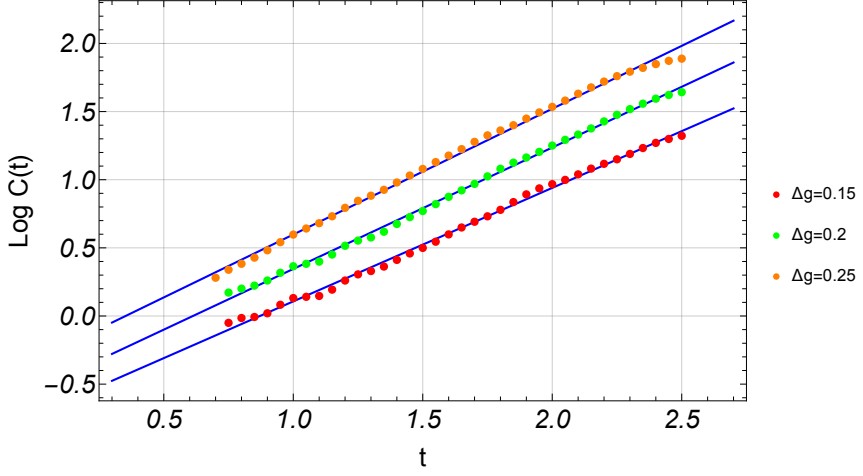

Figure 8: Plot of $Log(C(t))$ with time $t$ for different quench amplitude $\Delta g$.

## B   Details of tensor network for §2.3

The low lying initial state before the quench was achieved using Density Matrix Renormalization Group(DMRG) [42,43]. And the time-evolution in the OTOC was done using the recently

developed time-dependent variational principle (TDVP) [44–46]. The MPSs and Matrix Product Operators in the entire Tensor Network calculations have been constructed using ITensor C++ library (https://itensor.org).

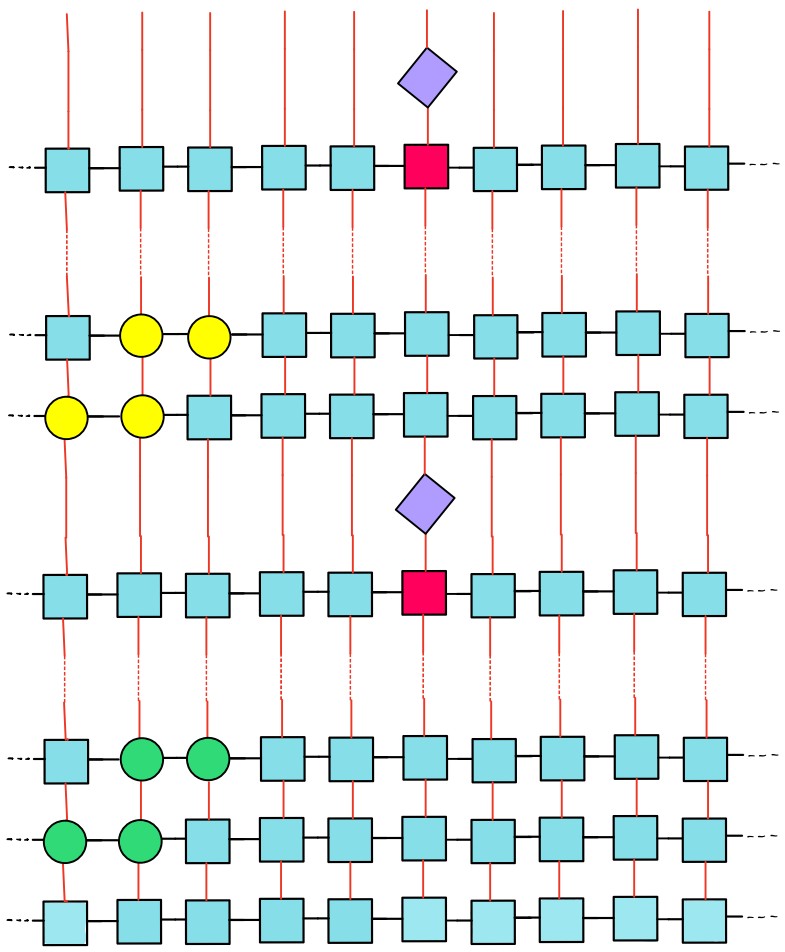

Figure 9: The green and yellow tensors denote the two-site centers of the mixed canonical form for the forward and backward time evolution respectively. This changes with sweeps left to right and right to left [49]. The sweeps on the network depends on the instantaneous Hamiltonian. Each layer depends on the instantaneous Hamiltonian. After time evolution to the desired time, the central site is made the one-site center of the mixed canonical form (red tensor) and acted on with $Z_{25}$ which is denoted by the violet tensor.

## B.1 DMRG

Here we use the strictly single site DMRG(DMRG3s) [47] from an initial random matrix product state(MPS) of bond-dimension $\chi = 10$, to approach a low lying state. This is then followed with an application of single-site DMRG [48]. We use DMRG3s initially as it avoids getting stuck in local minimas which is the case with single-site DMRG.

## B.2 TDVP

For an initial MPS state $|\Psi(A)\rangle$ the time evolution using TDVP can be understood as an orthogonal projection of the evolution vector of the Schrödinger equation onto the tangent space of

the present MPS manifold numerically restricted by dimension $\chi$

$$\frac{d|\Psi(A)\rangle}{dt} = -i\hat{\mathcal{P}}_{\mathcal{M}_{MPS_\chi}} \mathcal{H}|\Psi(A)\rangle,\tag{B.1}$$

where $\mathcal{P}_{\mathcal{M}_{MPS_\chi}}$ is the projector that maps $\Psi(A)$ to the tangent space of the MPS manifold of dimension $\chi$. We follow the prescription of [45, 49] to implement the one-site and 2-site TDVPs. For initial times until we reach the maximum bond dimension $\chi = 256$ we do the TDVP evoultion with 2-site TDVP as this helps in extending the bond dimension, this is shown in FIG. 9. If the maximum bond dimension is reached we switch to single-site TDVP (this is not expected as the time evolution of each Hamiltonian is only for short periods). Note that unlike the traditional use of TDVPs which evolve the state over a time $t$ with discrete steps $\tau$, we make an evolution over a very short period $\tau = 0.01$, then change the $g$ value in the Hamiltonian and make a short TDVP evolution again. This step is repeated until the desired value of time is reached. The value of $\tau$ is taken to be small not because of the error incurred as it has been previously shown that steps smaller that $\tau = 0.1$ with properly converged Lanczos exponentiation do not affect the evolution [50], but rather to have more OTOC values in the desired transient interval.

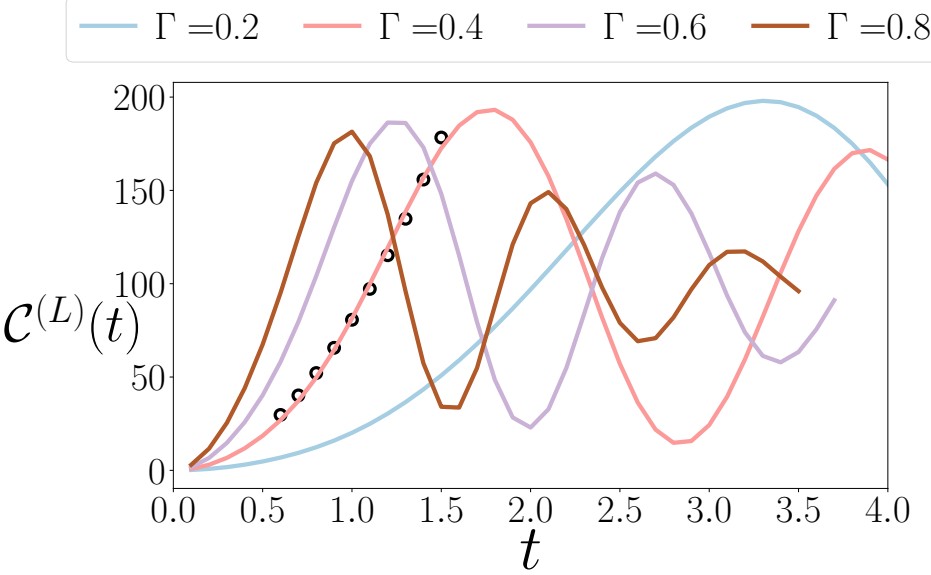

Figure 10: The commutator obtained from DOTOC ($c^{(L)}(t)$) for varying $\Gamma$'s. The black circles show a typical fit made in the increasing ramp regime.

## C  Density of the out-of-time-ordered correlator

Using the density of the out-of-time-ordered correlator (DOTOC) defining a commutator as

$$c^{(L)}(t) = -\langle [M_x(t), M_x(0)]^2\rangle/L,\tag{C.1}$$

where $L$ is the number of sites ( in our case to 50), $M_x = \sum_i \hat{S}_x^i$. In the language of [51] this is the case of the near-integrable model with composite observables. We use the same starting state with a similar quench as given in the main text. For smaller values of the rate, the growth

of $c^{(L)}(t)$ lasts longer than the case of the standard commutator, while for higher rates it seems to peak and then oscillates as shown in Fig. 10.

We make a fit of the for $c^{(L)}(t)$ as $\alpha t^\beta$ in the increasing ramp as shown by the black circles in Fig. 10. From here we extract a new Lyapunov exponent (not strictly an exponent in the traditional sense) with similar scales as that of energy given as $\tilde{\lambda} = \alpha^{1/\beta}$. These give a reasonable agreement with the scaling we found for the standard commutators from OTOCs as shown in Fig. 11

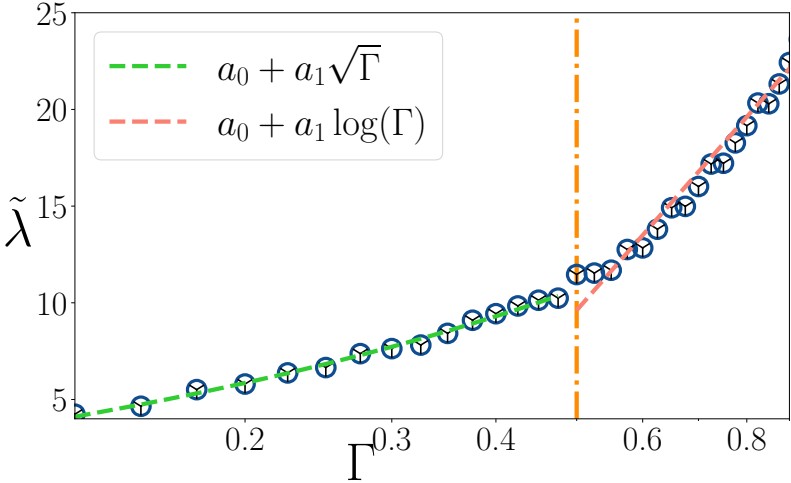

Figure 11: The new Lyapunov exponents extracted from the DOTOC commutators $\tilde{\lambda}$ is plotted with rate. The dashed lines represent the fit obtained with rate $\Gamma < 0.5$ and $\Gamma > 0.5$. While the dotted-dashed line differentiates $\Gamma = 0.5$.

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
