# Peer review of "Scrambling under quench"

_SciPost Physics Core, doi:SciPost Phys. Core 6, 021 (2023)_

## Round 1 · Referee Report · Anonymous (Referee 1) · 2022-7-9

Report

The paper computes out-of-time-order correlators in three systems after quantum quenches: (1) a system of coupled harmonic oscillators; (2) 2D CFT with large central charge; and (3) the quantum Ising model with longitudinal and transverse magnetic fields. It is claimed that Lyapunov exponents can be extracted for each system, and that they can be identified with the energy generated by the quench.

What appears problematic to me in the analysis of systems (1) and (3) is that no convincing evidence is provided that a meaningful Lyapunov exponent can be extracted, i.e., that the correlator really displays a period of exponential growth that lasts sufficiently long to distinguish it from e.g. power-law growth. If one finds a function that grows in a certain time range and fits it with an exponential, one will extract a best-fit `Lyapunov exponent’, but it will be meaningless if the function itself is e.g. a power rather than an exponential. For system (3), this issue was studied in detail in [47], where no exponential growth was found: the commutator squared stayed below the early-time power-law from the Baker-Campbell-Hausdorff formula. In fact, as discussed e.g. in 1803.05902, examples in which exponential growth has been found have been mostly limited to systems with large N or in semiclassical regimes. At the top of page 3, the authors acknowledge that for systems (1) and (3) “several e-foldings of exponential growth as observed in chaotic semi-classical systems are absent”, yet they do refer to the growth as exponential and claim to extract a Lyapunov exponent. To the best of my understanding, this is not meaningful.

Based on this, I unfortunately do not find the paper convincing enough to be published in SciPost.
  • validity: -
  • significance: -
  • originality: -
  • clarity: -
  • formatting: -
  • grammar: -

Author:  Diptarka Das  on 2022-10-07  [id 2899]

(in reply to Report 1 on 2022-07-09)
Category:
reply to objection
suggestion for further work

At the outset we will like to point out references $[2,18]$ in our manuscript where the physics of the Lyapunov index in QM models have been discussed. For the early times for which we as well as the authors of $[2,18]$ extract the Lyapunov index. We present in attached figure the logarithmic plot of $C(t)$ with time in early $(0.75-2.4)$ time regime and the logarithmic plot clearly shows that there are straight lines. This is a clear indication of exponential growth of $C(t)$ in the transient time regime. This is the time regime where we have fit our data for C(t) with $b \exp(2 \lambda t)$ to extract $\lambda$ for different quench amplitudes $\Delta g$. Please see attached figure for the plot of $Log(C(t))$ with time $t$ for different quench amplitude $\Delta g$.

With respect to example (3), it is our understanding that transient chaos is not ruled out. In fact the choice for our OTOC operators (non-local in terms of Jordan-Wigner fermions) were motivated by the fact, that they exhibit fast thermalization in quench set-ups (see Section 6.4.1 of this paper). Note that this is still in the integrable case. Even far away from the semi-classical regime, for a small integrability breaking parameter, transient chaos can still be present, see 1909.02145; and this is the regime in which we operate in example (3). Ultimately, the chaotic behaviour is tied to the presence of an effective temperature that is generated due to the quench. It was very important for us to work with large systems since such early time behaviour may easily be missed while working with smaller system sizes. This is also the reason we used the Tensor network techniques which by now is the standard tool for studying very early time dynamics. We suspect that 1908.08059 might have missed out on the transient chaos for spin $\tfrac{1}{2}$ due to finite size effects. Moreover the novelty of example (3) is the Kibble-Zurek (KZ) as well as fast quench scalings shown by the extracted Lyapunov exponent as a function of the quench rate $\Gamma$. It matches precisely the scalings shown by the quenched energy, and in fact also exhibits exactly the same transitions (from KZ $\sqrt{\Gamma}$ to fast $\log \Gamma$) at same values of $\Gamma$. For such scalings to exist it is required that we assume the commutator squared takes an exponential form with time. We tried fits of the form $a_1\exp(a_2 t)$ and $a_1 t^{a_2}$ \, i.e., between an exponential and polynomial curve. While we do find that the polynomial fit is marginally better with a standard deviation averaged over rates given as 0.03499931733146709, while for the exponential fit it is 0.09792172192525088, the polynomial fit has a very large exponent $a_2$ value of the order of ${\cal O}(20)$ for the transient times that we are interested in, again suggesting an exponential growth in transient time.

We completely agree with the referee that the scrambling regime is not present over sufficiently many e-foldings, and this is precisely because the scrambling time which is related to $\frac{1}{\lambda} \log \hbar$ is not that large. As the referee points out, this is also the reason why OTOC has been mostly studied analytically for large $N$ theories and in semiclassical limits, since the effective $\hbar \sim 1/N \ll 1$, and therefore the scrambling time is expected to last longer, and in these cases there is a connection with the classical Lyapunov exponent. Our intention however was not to connect the Lyapunov exponent $\lambda$ with the classical definition, rather as a quantity that gives the commutator growth dependency with energy density. In this context, it may also be important to take a note of recent results in this paper by Takeshi Morita wherein a single inverted harmonic oscillator has been studied as a prime example of exhibiting an interesting connection between classical Lyapunov exponent and quantum tunnelling. In this work by varying $\hbar$ the author explored how the distribution of excess energy due to quantum tunnelling takes a thermal form with temperature set by the classical Lyapunov exponent. Here clearly $\hbar$ is a parameter which is magnifies the thermality and may lead to interesting features if it continues to hold even in quench set ups.

Attachment:

t-vs-logC-new.pdf

---

## Round 1 · Referee Report · Anonymous (Referee 2) · 2022-7-15

Strengths

The authors study out-of-time-ordered correlators following quantum quenches in three different models and point out an interesting connection between the Lyapunov exponent as measured by OTOC and the amplitude of the quench. Interestingly, they also connect the scaling of the exponent to the eigenstate thermalisation hypothesis. While both ETH and chaotic behavior are associated to ergodicity, this is an interesting new connection and could be potentially fruitful to explore it further.

Weaknesses

As pointed out by the Referee A and the authors themselves, the models that the authors study only show quasi-exponential behavior at short times.

What are the variances of the obtained fit coefficients in Fig 2?

Report

The authors study out-of-time-ordered correlators following quantum quenches in three different models and point out an interesting connection between the Lyapunov exponent as measured by OTOC and the amplitude of the quench. Interestingly, they also connect the scaling of the exponent to the eigenstate thermalisation hypothesis. This could be an interesting direction to explore more in the future by the community. While I agree with the referee A that the authors only study systems with a transient growth of OTOC, I still think that it is an interesting new proposal and would recommend the article for a publication.

A few questions for the authors: - How tight is the bound that you showed? - Have you tried calculating quenches in any system with long time exponential growth of the OTOC? For example, how difficult would it be to generalise the exact solution in the SYK model to the quench scenario tat you are studying? (I'm not requesting the authors to do so for this article as this can be lengthy and hard problem. But I'm interested in the argumentation.) - What do you observe for the OTOC density from doi.org/10.1103/PhysRevB.96.060301 ? At least in the Ising case, this should not be too difficult with the DMRG methods that you have already implemented. There, you will of course observe algebraic and not exponential growth but maybe there is some interesting relation to the quench amplitude there as well.

Requested changes

  • Please report the variances of the fit coefficients from fig 2. It would make sense to plot only the region where the data points lie.

  • validity: good
  • significance: good
  • originality: high
  • clarity: top
  • formatting: excellent
  • grammar: excellent

Author:  Diptarka Das  on 2022-10-31  [id 2964]

(in reply to Report 2 on 2022-07-15)
Category:
remark
answer to question

We thank the referee for the very useful report.

What are the variances in Fig 2.

In Fig. 2 we fitted the extracted $\lambda$ with $a + c (\Delta g)\theta$. Upon including the variances, the coefficients lie in the following regions : $a \in [−0.04, 0.01], c \in [0.2, 0.25]$ and $\theta \in [0.25, 0.33]$. We will make sure to mention these ranges in our future resubmission.

We thank the referee for raising some interesting and open ended questions. Here we make an attempt to answer them:

How tight is the bound that you showed?

The bound assumes a power law behaviour of the density of states. More crucially we assume the validity of the ETH. Finally the bound follows from the MSS bound of chaos : $\lambda \leq 2\pi / \beta$. Thus the tightness depends on each of the factors. In this sense our bound may be improved by a lot for a specific system.

..quenches in SYK like models ...

This is one of the major future directions that we want to pursue. It has been shown that for SYK model, the system eventually thermalizes under quenching. To compute OTOC during quench in SYK [or in any other large $N$ models] we need to consider the Schwinger-Keldysh contour with four real time branches. This will give us a summation of ladder diagram which can be written in terms of a self-consistency equations, known as the Bethe-Salpeter equation, which need to be solved numerically. We expect that there too one can find a scaling between the Lyapunov exponent and the energy injected during quench. The main difficulty stems from the fact that instead of equilibrium Bethe-Salpeter equations, now these will be integro-differential ones [analogous to the Kadanoff-Baym equation] and hence will complicate the numerical analysis.

What do you observe for the OTOC density from doi.org/10.1103/PhysRevB.96.060301 ?

We are thankful for this suggestion. And with our adapted DMRG code we have the following observations to state: Using the density of the out-of-time-ordered correlator (DOTOC) defining a commutator as

$$ c^{(L)}(t)= -\langle [ M_x(t), M_x(0) ]^2 \rangle / L $$
where $N$ is the number of particles ( in our case to $50$) and $M_x = \sum_i \hat{S_x^i}$. In the language of PRB.96.060301 this is the case of the near-integrable model with composite observables. We use the same starting state with a similar quench as given in the main text. For smaller values of the rate, the growth of $c^{(L)}(t)$ lasts longer than the case of the standard commutator, while for higher rates it seems to peak and then oscillates as shown in the first figure in the attached file DOTOC-Figs.pdf. We make a fit of the for $c^{(L)}(t)$ as $\alpha t^{\beta}$ in the increasing ramp as shown by the black circles. From here we extract a new Lyapunov exponent (not strictly an exponent in the traditional sense) with similar scales as that of energy given as $\tilde{\lambda} = \alpha^{1/\beta}$. These give a reasonable agreement with the scalings (both Kibble-Zurek, as well as fast scaling) that we found for the standard commutators from OTOCs as shown in the second figure in the attached file.

Attachment:

DOTOC-Figs.pdf

---

## Round 1 · Referee Report · Anonymous (Referee 3) · 2022-8-26

Report

The authors study the possible exponential growth of the OTOC for several models, including sudden quenches for 2 coupled harmonic oscillators, large central charge CFTs and smooth quenches in transverse field Ising models. While this is a collection of interesting results, the main motivation remains somewhat blurry and I could not identify the main message behind all these results. Most importantly why not study sudden and smooth quenches for all models studies and contrast/parallel their behaviour? Especially for the coupled HO, this should be rather straightforward.

All in all, I feel that since the main convincing message is lacking, the paper is suitable to Scipost Physics Core only.

Requested changes

  1. Investigate sudden AND smooth quenches for all models or explain why it is not necessary.

  2. The figure captions are also rather poor, by e.g. looking at Fig. 6, it is not obvious which model it corresponds to, what the colors mean etc. The figures should be made more self-contained.

  3. The Kibble-Zurek scenario is not explained at all, some more emphasis should be put on this as well.

  4. I had also problems with understanding the conclusions section, some streamlining would be beneficial for the general reader.

  • validity: -
  • significance: -
  • originality: -
  • clarity: -
  • formatting: -
  • grammar: -

Author:  Diptarka Das  on 2022-10-31  [id 2965]

(in reply to Report 3 on 2022-08-26)

We thank the referee for their comments and suggestions, to which we respond below:

Investigate sudden AND smooth quenches for all models or explain why it is not necessary.

In case of the coupled quantum Harmonic oscillators, the exercise for smooth quenches is computationally very intensive. This will require us to solve for the instantaneous eigenfunctions and values by solving the Laplacian for every instant of time, each with a different coupling. This increases the computational complexity $T/\Delta T$ times in comparison with the sudden quench problem [which needs only the eigensystem of the initial and the final Hamiltonia], where $T$ is the time for which we evolve the system and $\Delta T$ is the timescales that we resolve numerically. Besides this technical hurdle, a main point that we elucidate with the oscillator example is the reflection of thermalization in scrambling. For this point, we do not need to quench smoothly. The sudden quench set-up is enough in this example to establish that the Lyapunov exponent is related to the quench energy. This naturally defines an effective temperature (See our Fig 3.), which is consistent with earlier thermal oscillator results of 2004.04381.

In the two dimensional CFT case, sudden quench was studied because unlike the Cady-Calabrese proposal there is no well-defined prescription to approximate states created by smooth quenches. A starting guess maybe to write down a variational ansatz similar to 1706.01568 and trying to fix the parameters (which will now be time-dependent) by minimizing the instantaneous energy. A more conservative approach will be to use conformal perturbation theory. For one point functions this has worked out by one of us --- See Appendix A of 1706.02322. However for the OTOC computation this naturally leads to very high point functions (that need to be suitably integrated), which our bCFT set-up is specially designed to avoid, and will not lead to analytically tractable late time answers.

In the case of the Ising model, our focus was not on thermalization but on exhibiting the universal scalings associated with smooth quenches. Unlike the previous two examples, this model being amenable to sophisticated 1D numerical methods, allowed us to study smooth quenches : and indeed find both the Kibble-Zurek as well as the fast scalings. Interestingly even the algebraic DOTOC shows these scaling behaviours [please refer to our reply to referee 2]. Furthermore, the sudden quench for the Ising case is not interesting enough in the context we are interested in to report. It had been investigated earlier while being proposed as a non-equilibrium order parameter in this paper and had also been investigated in one of ours Masters thesis.

The figure captions are also rather poor ...

We thank the referee for noticing the incompleteness in Fig. 6. Here we modify the figure and the caption shown in the attachment Fig6.pdf. We believe other figures with captions are clearly mentioned.

The Kibble-Zurek scenario is not explained at all...

We thank the referee for noticing this shortcoming. We will gladly refer in detail to the excellent review of KZ as well as other recent scalings from this review in our future resubmission.

I had also problems with understanding the conclusions section, some streamlining would be beneficial for the general reader.

We will streamline the conclusions in our future resubmission.

We hope that we have addressed the main concerns of referee 3. We also remain convinced, resonating the viewpoint of the 2nd referee that this is an interesting new connection and could be potentially fruitful to explore. Thus, we will like to request referee 3, to recommend publication in SciPost and not in SciPost Physics Core.

Attachment:

Fig6.pdf

---

## Round 2 · Referee Report · Anonymous (Referee 5) · 2022-12-19

Report

The authors have addressed my comments satisfactorily. The paper hits the standard of Scipost Physics Core, therefore I recommend its publication.

---

## Round 2 · Referee Report · Anonymous (Referee 4) · 2022-12-19

Report

I'm happy with the authors' modifications and additional explanations. I recommend the manuscript for publication.

---

## Round 2 · Author Response

Implementing suggested changes from the referees we have updated our draft.

---

## Round 2 · List of Changes

All the changes are marked in color blue:

1. The introduction now has a paragraph on page 3 with comments on why we study sudden and smooth quenches for the cases we do. This is to address point 1 of anonymous Referee 3 of SciPost during our previous submission.

2. Appendix A.1 has been added along with Fig. 8 which clarifies the extraction of the exponent. This is to address partially the objections of anonymous Referee 1 of SciPost during our previous submission.

3. Captions in Figures have been improved. In particular variances have been added in the plot for Fig. 2 implementing the change suggested by anonymous Referee 2 of SciPost during our previous submission. Fig. 6. caption has also been improved taking into account the suggestion of anonymous Referee 3.

4. In order to address the point 3 of anonymous Referee 3 we have included a review of Kibble-Zurek and fast scaling during quenches [1] as well as added some explanations of the scalings in section 2.3.

5. Following the suggestion of Referee 2 we have added a new Appendix C where we analyze the algebraic growth in DOTOC and explore the scalings in the corresponding rate controlling the growths. This appendix also contains two new figures, 10 and 11.

6. We have streamlined the Conclusions section in the updated submission. This is to address point 4 of anonymous Referee 3 of SciPost during our previous submission.

7. We added a line in our Acknowledgements section thanking the Referee 2 for urging us to investigate DOTOC, leading to discovering also universal scalings in the extensive quantity.

---

## Editorial Decision

published